# Antimycobacterial and Nitric Oxide Production Inhibitory Activities of Triterpenes and Alkaloids from *Psychotria nuda* (Cham. & Schltdl.) Wawra

**DOI:** 10.3390/molecules24061026

**Published:** 2019-03-15

**Authors:** Almir Ribeiro de Carvalho Junior, Rafaela Oliveira Ferreira, Michel de Souza Passos, Samyra Imad da Silva Boeno, Lorena de Lima Glória das Virgens, Thatiana Lopes Biá Ventura, Sanderson Dias Calixto, Elena Lassounskaia, Mario Geraldo de Carvalho, Raimundo Braz-Filho, Ivo Jose Curcino Vieira

**Affiliations:** 1Instituto Federal de Santa Catarina, Câmpus Criciúma, Criciúma 88800-000, Santa Catarina, Brazil; 2Colegiado de Ciências Exatas e Biotecnológicas, Universidade Federal do Tocantins, Gurupi 77400-000, Tocantins, Brazil; rafaellaoliveira@mail.uft.edu.br; 3Laboratório de Ciências Químicas, Centro de Ciência e Tecnologia, Universidade Estadual do Norte Fluminense Darcy Ribeiro, Campos dos Goytacazes, Rio de Janeiro 20000-000, Brazil; michelpassos19@gmail.com (M.d.S.P.); smr.imad@gmail.com (S.I.d.S.B.); lorena_limagloria@hotmail.com (L.d.L.G.d.V.); brazfilhor@gmail.com (R.B.-F.); curcinovieira@gmail.com (I.J.C.V.); 4Laboratório de Produtos Bioativos/LPBio, Curso de Farmácia, Universidade Federal do Rio de Janeiro, Campus Macaé, Macaé, Rio de Janeiro 20000-000, Brazil; thativentura@yahoo.com.br; 5Laboratório de Biologia do Reconhecer, Universidade Estadual do Norte Fluminense Darcy Ribeiro, Campos dos Goytacazes, 28013-602, Rio de Janeiro 20000-000, Brazil; sandersoncalixto@yahoo.com.br (S.D.C.); elena@uenf.br (E.L.); 6Departamento de Química, Instituto de Ciências Exatas, Universidade Federal Rural do Rio de Janeiro, Seropédica, Rio de Janeiro 20000-000, Brazil; mgeraldo@ufrrj.br

**Keywords:** Rubiaceae, *Psychotria nuda*, alkaloids, Mycobacterium, inflammation, nitric oxide

## Abstract

A phytochemical study of leaves and twigs of *Psychotria nuda* resulted in 19 compounds, including five indole alkaloids, *N*,*N*,*N*-trimethyltryptamine, lyaloside, strictosamide, strictosidine, and 5α-carboxystrictosidine; two flavonolignans, cinchonain Ia and cinchonain Ib; an iridoid, roseoside; a sugar, lawsofructose; a coumarin, scopoletin; a diterpene, phytol; three triterpenes, pomolic acid, spinosic acid, and rotungenic acid; and five steroids, sitosterol, stigmasterol, campesterol, *β*-sitosterol-3-*O*-*β*-d-glucoside, and *β*-stigmasterol-3-*O*-*β*-d-glucoside. Some compounds were evaluated for their in vitro activity against *Mycobacterium tuberculosis* and their ability to inhibit NO production by macrophages stimulated by lipopolysaccharide (LPS). The compounds pomolic acid, spinosic acid, strictosidine, and 5α-carboxystrictosidine displayed antimycobacterial activity with minimum inhibitory concentrations ranging from 7.1 to 19.2 µg/mL. These compounds showed promising inhibitory activity against NO production (IC_50_ 3.22 to 25.5 μg/mL). *5*α-carboxystrictosidine did not show cytotoxicity against macrophages RAW264.7 up to a concentration of 100 µg/mL. With the exception of strictosamide, this is the first report of the occurrence of these substances in *P. nuda*.

## 1. Introduction

Tuberculosis (TB) has become the communicable disease with the highest mortality rate around the world and the main cause of death among people living with HIV. Estimates point out that in 2016, 10.4 million people became ill and 1.7 million died [1,2]. The situation is further complicated by the emergence of multidrug-resistant (MDR) and extensively drug-resistant (XDR) TB and by HIV/AIDS coinfection [3,4]. Thus, development of new anti-tubercular drugs is essential for curing the disease. Many diseases such as cancer, Alzheimer’s disease, and atherosclerosis, as well as infections such as TB are followed by inflammatory processes with high production of chemical mediators [5]. In general, for protection against mycobacteria, the production of proinflammatory mediators by the infected macrophages is essential. However, in cases of severe forms of TB, additional anti-inflammatory therapy is required to prevent excessive inflammation [6,7].

The genus *Psychotria* (Rubiaceae) comprises approximately 2000 species occurring mostly in tropical and subtropical regions of the world [8]. Some of its species are widely used in folk medicine for purposes such as earache [9], abdominal pain [10], constipation [11], coughs [12], etc. *Psychotria* species have been studied for their diversity of secondary metabolites such as alkaloids [13], flavonoids [14], triterpenes [15], coumarins [12], and iridoids [14], and stand out for their antimycobacterial [16], anti-inflammatory [17], cytotoxic [15], analgesic [18], and antimicrobial [19] activities.

This study aims to investigate the chemical profile of *Psychotria nuda* methanol extract collected in the Biological Reserve of Poço das Antas (Nova Iguaçu, RJ, Brazil). In addition, due to the previously described antimicrobial activity of *P. nuda* [20], the antimycobacterial activity of the compounds isolated in this chemical study and their ability to inhibit LPS-stimulated NO production in macrophages were also investigated.

## 2. Results and Discussion 

### 2.1. Chemical Study

The chromatographic fractionation of leaf and twig extracts of *P. nuda* led to the isolation of 19 compounds: pomolic acid (**1**), spinosic acid (**2**), sitosterol (**3**), stigmasterol (**4**), campesterol (**5**), phytol (**6**), *β*-sitosterol-3-*O*-β-d-glucoside (**7**), *β*-stigmasterol-3-*O*-*β*-d-glucoside (**8**), cinchonain Ia (**9**), cinchonain Ib (**10**), *N*,*N*,*N*-trimethyltryptamine (**11**), lyaloside (**12**), lawsofructose (**13**), roseoside (**14**), strictosamide (**15**), scopoletin (**16**), rotungenic acid (**17**), strictosidine (**18**), and *5*α-carboxystrictosidine (**19**) (Figure 1). Substance identification was based on spectral analysis and data comparison with values described in the literature [21,22,23,24,25,26,27,28,29,30,31,32,33,34]. This is the first report of alkaloid (**11**) isolation from a natural product, as well as the isolation of the iridoid (**14**) and the flavonolignans (**9**) and (**10**) from the genus *Psychotria*.

Compound **11** was isolated from the twigs of *P. nuda* as a brown oil. The HRESIMS (Appendix A) revealed a molecular formula of [C_13_H_19_N_2_] of *m*/*z* 203.1501 (calculated 203.1548). The IR spectrum displayed broad, strong absorption at 3410 cm^−1^, characteristic of the N-H vibration in the indole moiety as well as other signals. The ^1^H-NMR spectrum (Table 1) showed the typical aromatic resonance pattern for 3-substituted indole alkaloids. Thus, there was the ABCD system of the aromatic ring [δ_H_ 7.64 (d, *J* = 8.0 Hz), 7.09 (t, *J* = 7.5 Hz), 7.15 (t, *J* = 7.5 Hz), 7.41 (d, *J* = 8.0 Hz)], and the signal at δ_H_ 7.25 (H-2). The side chain attached at C-3 of the indole unit consisted of two methylene groups and three *N*-methyl groups. The methylene at δ_H_ 3.27 displayed HMBC correlations to C-2 (δ_C_ 122.9), C-3 (δ_C_ 107.9), and C-4 (δ_C_ 126.6), showing that it was directly attached to the indole at C-3. It also had a COSY correlation to the methylene at δ_H_ 3.62, which had HMBC correlations to the *N*-methyl groups (δ_H_ 52.2). Thus, the structure of **11** was established as *N*,*N*,*N*-trimethyltryptamine. The detailed MS analysis (Figure 2) was used to confirm the structure of **11**. This is the first report of the isolation of this alkaloid from a natural product; its detection by HPLC-MS was previously reported in citrus extracts [35].

*Psychotria* species are bioproducers of tryptamine and β-carbolinic alkaloids; the indole skeleton of these alkaloids has a structural correlation with serotonin (5-HT), which may be related to the neurological effects attributed to these alkaloids from interaction with serotonergic receptors [36]. The alkaloids are the main active constituents of ayahuasca tea. Ayahuasca is a psychotropic drink from South America initially used in rituals of indigenous tribes of the Amazon region. The preparation of Ayahuasca consists of the firing of *Banisteriopsis caapi* (Malphighiaceae) stems together with leaves of *Psychotria viridis*. Tea consumption increases serotonin concentrations and makes oral dimethyltryptamine bioavailable, provoking a hallucinogenic reaction [36,37]. The identification of the indolic alkaloids **11**, **15**, **18**, and **19** in extracts of *P*. *nuda* suggests the psychotropic potential of this species.

### 2.2. Antimycobacterial Activity

In a previous study from our group [20], the ethanolic extract from leaves of *P. nuda* showed antimycobacterial activity (MIC_50_ 8.32 ± 2.39 µg/mL) against *Mycobacterium bovis* BCG. In this work, a mixture of triterpenes (**1** + **2**) and two alkaloids (**18** and **19**) was isolated from this species and evaluated against *Mycobacterium tuberculosis* H37Rv and the hypervirulent strain M299 (Table 2).

All tested substances showed activity against *M. tuberculosis* H37Rv. The most active compound was strictosidine (**18**, MIC_50_ 7.1 µg/mL), followed by a mixture of pomolic and spinosic acids (**1** + **2**, MIC_50_ 19.2 µg/mL) and 5-α-carboxystrictosidine (**19**, MIC_50_ 26.3 µg/mL). Although there are reports on the antimycobacterial activity of the alkaloid strictosidine (MIC > 50 µg/mL) in the literature [38], it was re-evaluated against H37Rv as well as against the hypervirulent strain M299. The mixture of triterpenes (**1** + **2**) and alkaloid **19** were inactive against the hypervirulent strain M299 at the maximum tested concentration of 100 µg/mL.

Plant-derived terpenoids have shown moderate to significant biological activity against *M. tuberculosis* [39]. The common triterpenes—oleanolic acid, ursolic acid, and betulinic acid—and their respective hydroxymethyl analogues—erythrodiol, uvaol, and betulin—all showed similar activity, with minimal inhibitory concentration (MIC) values of 32 or 64 µM. Some studies have shown that the structural class of pentacyclic triterpene and the stereochemistry of the hydroxy group at C-3 appear to be less important for antimycobacterial activity. However, oxidation of the C-3 hydroxy group of lupeol led to loss of activity for lupeol acetate and lupenone. Pomolic acid (**1**) isolated from *Acaena pinnatifida* showed moderate activity against *M. tuberculosis* (MIC 64 μM) [40]. In this work, we report the antituberculosis action of this triterpene (**1**) when in a mixture with spinosic acid (**2**) with an MIC of 19.2 ± 0.2 µg/mL. The mixture of triterpenes (**1** + **2**), despite having an acceptable inhibitory effect on virulent, were highly toxic according to the MTT assay (Table 3).

The antimycobacterial activity of indole alkaloids has already been reported by many authors. Globospiramine, a novel spirobisindole alkaloid isolated from *Voacang globosa* (Apocynaceae), showed promising activity against *M. tuberculosis* H37Rv (MIC 4.0 µg/mL). The other structural analogues tested showed an MIC > 50 µg/mL. Considering the structure–activity relationship, the authors suggested the importance of the presence of the hydroxyl group in C-3 and the absence of C-2 oxidation in the inhibitory activity of globospiramine [41]. Bioassays performed with a bioactive fraction of *Alstonia scholaris* (Apocynaceae) have shown that strychnane indole alkaloids such as tubotaiwine (MIC 100 µg/mL) are more active than the vallesamine–apparicine subtype indole alkaloids, showing that the monoterpenoid skeleton of indole alkaloids influences their activity against *M. tuberculosis* H37Rv [42]. In this paper, we report the promising antituberculosis activity of the monoterpene indole alkaloids strictosidine (**18**) and 5-α-carboxy-strictosidine (**19**). The lipophilicity of the substances is considered an important factor for antimycobacterial activity [43,44]. The presence of the carboxyl at the C-5 position of substance **19** increases its polarity with respect to strictosidine (**18**), which may be related to its lower inhibitory activity against *M. tuberculosis* H37Rv and the hypervirulent strain M299. These results suggest that these substances may contribute to the antimycobacterial activity observed for the extracts of *P. nuda* [20].

### 2.3. Inhibition by Triterpenes and Alkaloids of LPS-induced NO Production and Cytotoxicity in RAW264.7 Cells 

Plants or their extracts belonging to the genus *Psychotria* have been used in folk medicine in the treatment of inflammation [45]. Indole alkaloids, the main chemosystematic markers of the genus *Psychotria* [13], also exhibited anti-inflammatory activities in vitro [46] and in vivo [47]. These data suggest that both extracts of these species and indole alkaloids could serve as models for the development of new anti-inflammatory drugs. In this study, the anti-inflammatory properties of alkaloids and triterpenes were tested in LPS-induced RAW264.7 cells.

Inflammatory mediators such as NO and PGE2 play crucial roles in the pathogenesis of inflammatory diseases. The inhibition of these inflammatory mediators is regarded as a therapeutic method for the prevention of inflammation [48]. The results showed that all the compounds studied act as potent inhibitors of NO (with IC_50_ values below 26 µg/mL). Of note are the compounds **18** and **19**, which, besides being strong anti-inflammatory agents via inhibition of NO production, also have almost no cytotoxic effects (Table 3).

A cell proliferation assay was performed to ensure that inhibition of NO production was not related to cytotoxic effects. The effects of triterpenes (**1** + **2**) and alkaloids (**18** and **19**) on cell viability were determined by MTT assay on RAW264.7 cells (Table 3). A mixture of pomolic acid (**1**), spinosic acid (**2**), and strictosidine (**18**) reduced cell viability significantly at 20 µg/mL (*** *p* < 0.001 and * *p* < 0.05, respectively) in the presence of LPS. Cell viability was not affected by *5*α-carboxystrictosidine (**19**) in RAW264.7 cells up to 100 µg/mL (data not shown). Based on the results obtained in this study, we suggest that alkaloids **18** and **19** may serve as templates for new anti-inflammatory drug development.

Some studies describe the anti-inflammatory action of indole alkaloids. The 6-hydroxy-3,4-dihydro-1-oxo-β-carboline and hippophamide indole alkaloids showed potent inhibitory activities against NO production in LPS-treated RAW264.7 macrophages, with IC_50_ values of 31.92 ± 0.01 and 25.16 ± 0.41 μM, respectively, and did not show cytotoxicity at the inhibitory concentration. Structure–activity relationship studies (SAR) have shown that the lactam moiety of these alkaloids contributes more to their anti-inflammatory activity than the indole group [46]. Strictosamide (**15**) isolated from *Nauclea officinalis* showed in vivo anti-inflammatory activity against 12-O-tetradecanoylforbol-13-acetate (TPA)-induced ear oedema in mice, as well as vascular permeability and leukocyte migration induced by carboxymethylcellulose sodium [47]. Thus, the indolic alkaloids **15**, **18**, and **19** described in this study may contribute to the anti-inflammatory activity reported for extracts of *P. nuda* [20].

## 3. Materials and Methods

### 3.1. General Experimental Procedures

FTIR spectra were recorded on an IRAffinity-1 Shimadzu spectrometer (Quioto, Kansai, Japan) using a KBr disk. The NMR analyses were carried out on a Bruker Ascend 500 (Billerica, MA, USA) in pyridine-d_5_, CDCl_3_, or methanol-d_4_ at 500 MHz for ^1^H and 125 MHz for ^13^C using TMS as an internal reference. Chemical shifts (δ) are expressed in ppm and coupling constants (J) in Hz. HR-ESI-MS mass spectra were obtained on a micrOTOF-Q II Bruker Daltonics mass spectrometer (Billerica, MA, USA) using positive and negative ion modes of analysis. Gas chromatography coupled with mass spectrometry (GC/MS) of low-resolution experiments was carried out on a GCMS-QP5050A Shimadzu operating with an ionization energy of 70 eV. Column chromatography (CC) was performed using silica gel 60 (Merck, Darmstadt, Alemanha) (0.063–0.200 mm) and Sephadex™ LH-20 (Sigma-Aldrich, St. Louis, MO, USA). Silica gel F254 was used for preparative thin-layer chromatography (TLC); aluminum-backed Sorbent silica gel plates, w/UV 254, were used for analytical TLC (Merck) with visualization under UV (254 and 366 nm), vanillin, and iodine vapor.

### 3.2. Plant Material

Leaves and twigs of *P. nuda* were collected at the Reserva Biológica de Poço das Antas, Nova Iguaçu-RJ, Brazil and identified by the botanist Sebastião José da Silva Neto. Both voucher specimens (H9726) are deposited at the herbarium of UENF.

### 3.3. Preparation and Fractionation of Methanol Extract

Air-dried powdered twigs of *P. nuda* (1.45 kg) were extracted exhaustively with methanol at room temperature, affording 36.7 g of crude extract. After suspension in a MeOH:H_2_O (1:3) solution, part of this extract (36.0 g) was partitioned with dichloromethane, ethyl acetate, and n-butanol. The dichloromethane fraction (3.0 g) was subjected to repeated silica gel CC using CH_2_Cl_2_ and CH_2_Cl_2_:MeOH, leading to the identification of a mixture of compounds **3**-**5** (153.0 mg), **6** (23.2 mg), and a mixture of **7** and **8** (53.0 mg). The ethyl acetate fraction of twigs (790.0 mg) was similarly chromatographed (except for the use of CH_2_Cl_2_:AcOEt as eluent), allowing the identification of compounds **9** and **10** (35.0 mg). The n-butanol fraction (1.5 g) was purified by CC (CH_2_Cl_2_ and CH2Cl2:MeOH as eluent), yielding 35 fractions. Fraction 29 (175 mg) after purification on Sephadex LH-20 led to the identification of compound **11** (26.0 mg).

The n-butanol fraction of leaves (1.85 g) was chromatographed, leading to the identification of compounds **12** and **13** (in a mixture, 64.2 mg). Another fraction (88.0 mg) obtained from this procedure was purified on Sephadex LH-20, and compounds **14** (13.0 mg) and **15** (11.0 mg) were identified. The ethyl acetate fraction (920 mg) was similarly chromatographed, affording 18 fractions. Fraction 5 (28.3 mg) was purified by preparative TLC, and compound **16** (4.0 mg) was obtained. Fraction 12 (114.5 mg) was rechromatographed, leading to the identification of a mixture of compounds **1** and **2** (21.0 mg). Fraction 16 (121.3 mg) was rechromatographed, leading to the isolation of compound **17** (26.0 mg). The methanolic fraction of leaves (1.7 g) was fractionated by CC on silica gel and eluted with CH_2_Cl_2_ and CH_2_Cl_2_:MeOH solutions (up to 20% MeOH), affording 12 fractions. Fraction 5 (200 mg) was similarly chromatographed, and after purification by CC on Sephadex LH-20 (eluted with MeOH), compound **18** (22.0 mg) was identified. Analogously, compound **19** (32 mg) was obtained from Fraction 10 (210 mg). All substances were submitted to ¹H and ^1^³C nuclear magnetic resonance analysis, and substance 11 was additionally identified with infrared and HR-ESI-MS.

### 3.4. Antimycobacterial Activity

Two strains of mycobacteria evaluated for virulence in a previous study [49] were used in this study—the virulent *M. tuberculosis* laboratory strain H37Rv (ATCC 27294) and the highly virulent *M. tuberculosis* strain Beijing M299 isolated from a TB patient in Mozambique.

All samples were evaluated for their antimycobacterial activity at concentrations of 0.8, 4, 20, and 100 µg/mL using an MTT assay [50]. *M. tuberculosis* H37Rv (ATCC 27294) and the hypervirulent strain Beijing M299 were incubated with Middlebrook 7H9 medium supplemented with 0.05% glycerol, 0.05% Tween 80, and ADC (albumin dextrose catalase). Cells were plated at a density of 1 × 106 CFU/well in a 96-well plate and treated with the samples. The plate was incubated at 37 °C and 5% CO_2_ for 5 days. Posteriorly, the MTT solution was incubated for 3 h, and the lysis buffer was added [20% *w*/*v* sodium dodecyl sulphate (SDS) and 50% dimethylformamide (DMF) in distilled water, pH 4.7]. The plate was incubated overnight, and the reading was carried out using a spectrophotometer at 570 nm. As a positive control, *M. tuberculosis* treated with the antibiotic rifampicin (95% purity; Sigma-Aldrich) was used (ranging from 0.00032 to 1 μg/mL for M. tuberculosis H37Rv and 0.008 to 10 μg/mL for the *M. tuberculosis* M2b clinical isolate). As a negative control, untreated *M. tuberculosis* was used.

### 3.5. Determination of Nitric Oxide Production by RAW 264.7 Macrophages

The murine macrophage cell line RAW 264.7 was obtained from the American Type Culture Collection (ATCC) and grown at 37 °C and 5% CO_2_ in DMEM F-12 supplemented with 10% FCS and gentamicin (50 µg/mL). Macrophages (1 × 105 cells/well) were seeded in 96-well tissue culture plates in the presence or absence of four concentrations of the samples (0.8, 4, 20, and 100 µg/mL) and/or LPS (*Escherichia coli* 055:B5; Sigma-Aldrich). After 24-h incubation, supernatants were collected, and the concentration of nitrite was determined according to the Griess test [51]. NG-methyl-L-arginine acetate salt (L-NMMA, Sigma-Aldrich, 98% purity) was used as a positive control.

### 3.6. Cytotoxic Effect

Cytotoxicity was assessed by the mitochondrial-respiration-dependent MTT reduction method [52]. RAW264.7 cells (5 × 105 cells/mL) in 96-well plates were incubated with increasing doses of the test compound (0.8, 4.0, 20, and 100 µg/mL of compounds or L-NMMA) at 37 °C in 5% CO_2_ for 24 h. After treatment, 5 µL MTT solution (5 mg/mL) was added to each well. After incubation for 2 h at 37 °C, the formazan crystals in viable cells were solubilized in HCl (4 mM) otisopropanol. The absorbance of each well was then read at 570 nm. The optical density of formazan formed in control (untreated) cells was taken as 100% viability.

### 3.7. Statistical Analysis 

The results obtained were tabulated by the LabChart 7 program and analyzed statistically through the software GraphPad Prisma 4. The tests were performed in three replicates, and values were expressed as mean ± SD.

## 4. Conclusions

Except for the alkaloid (**15**), this is the first report of the isolation of secondary metabolites from *P. nuda* and resulted in the identification of **19** substances, four of them unpublished in the genus Psychotria: flavonoligananas cinchonain Ia (**9**), cinchonain Ib (**10**), *N*,*N*,*N*-trimethyltryptamine alkaloid (**11**), and roseoside iridoid (**14**).

In addition, we report the promising antimycobacterial and NO production inhibitory activities of the triterpenes pomolic acid (**1**) and spinosic acid (**2**) and the alkaloids strictosidine (**18**) and *5*α-carboxystrictosidine (**19**). These results suggest that *P. nuda* is a new natural source of bioactive substances that may be of future utility against diseases such as TB and other inflammatory-related processes.

## Figures and Tables

**Figure 1 molecules-24-01026-f001:**
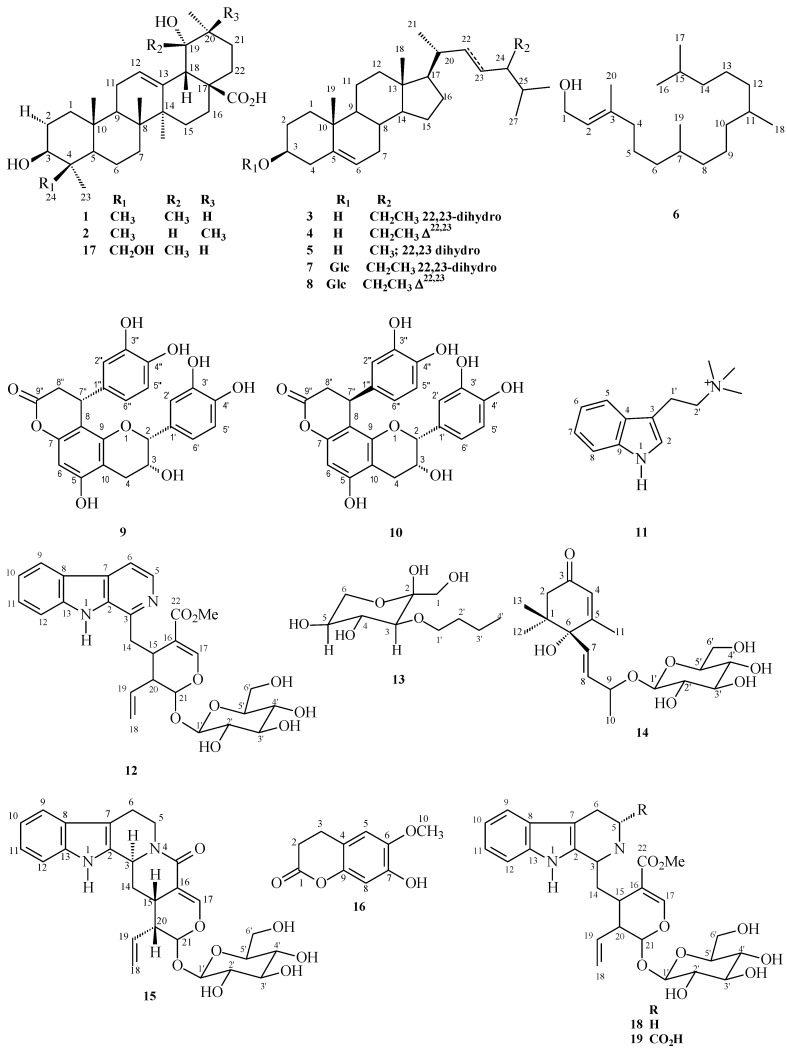
Structures of compounds **1**–**19**.

**Figure 2 molecules-24-01026-f002:**
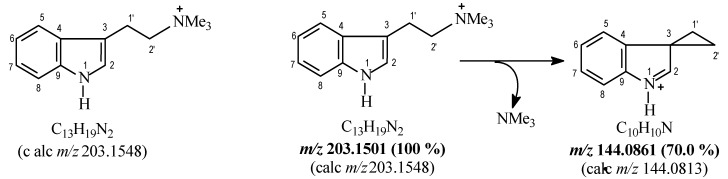
Proposed fragmentation to peaks detected in the mass spectrum of compound **11**.

**Table 1 molecules-24-01026-t001:** ^1^H-NMR (500 MHz) and ^13^C-NMR (125 MHz) spectral data for **11**, including results of HSQC and HMBC experiments. Chemical shifts δ in ppm and coupling constants in Hz.

	11				Literature *
	HSQC		HMBC		
**C**	δ_C_	δ_H_	^2^ *J* _CH_	^3^ *J* _CH_	δ_C_ **
3	107.97	-	2H-1′; H-2	2H-2′; H-5	110.8
4	126.65	-		2H-1′; H-2; H-6; H-8	127.0
9	136.78	-		H-2; H-5; H-7	136.5
**CH**					
2	122.95	7.25 (*s*)		2H-1′	123.8
5	118.77	7.64 (*d*, 8.0 Hz)		H-7	118.9
6	117.53	7.09 (*t*, 7.5 Hz)		H-8	118.7
7	121.44	7.15 (*t*, 7.5 Hz)		H-5	121.7
8	111.20	7.41 (*d*, 8.0 Hz)		H-6	111.9
**CH_2_**					
1′	18.95	3.27 (*m*)	2H-2′		19.0
2′	66.47	3.62 (*m*)	2H-1′	NMe_3_	65.5
**CH_3_**					-
N-M_3_	52.27	3.23 (*s*)			52.5

* [26]; ** DMSO-d_6_.

**Table 2 molecules-24-01026-t002:** Minimal inhibitory concentration (MIC) of triterpenes (**1** + **2**) and alkaloids (**18** and **19**), isolated from twigs and leaves of *Psychotria nuda* against *Mycobacterium tuberculosis* strains.

Substances	MIC (µg/mL)	
	H37Rv	M299
Pomolic acid (**1**) and spinosic acid (**2**)	19.2 ± 0.2	Inactive at 100 µg/mL
Strictosidine (**18**)	7.1 ± 0.6	33.1 ± 0.8
5α-carboxystrictosidine (**19**)	26.3 ± 1.9	Inactive at 100 µg/mL
Rifampicin ^1^	0.2 ± 0.1	1.1 ± 0.1

^1^ Standard anti-tuberculosis (TB) drug.

**Table 3 molecules-24-01026-t003:** Expression of IC_50_ for inhibition of NO production by stimulated macrophages and cytotoxicity in the MTT assay.

	IC_50_ (µg/mL)	
Compound	NO	MTT
**1 + 2**	25.5 ± 0.1	13.17±0.1
**18**	3.22 ± 0.1	90.7±0.1
**19**	3.44 ± 0.1	>500
**L-NMMA** ^1^	78.3 ± 6.5	>100

^1^ Standard inhibitor of NO.

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
