# Peer review of "Antimycobacterial and Nitric Oxide Production Inhibitory Activities of Triterpenes and Alkaloids from Psychotria nuda (Cham. & Schltdl.) Wawra"

_molecules, 2019, doi:10.3390/molecules24061026_

Round 1
Reviewer 1 Report
The manuscript by Junior et al. reports the significant anti-Mycobacterium tuberculosis activity of some compounds isolated from Psychotria nuda. Moreover, two of these compounds were shown to inhibit NO production by macrophages stimulated by LPS and revealed low cytotoxicity in RAW264.7 cells, being therefore promising anti-inflammatory agents.
This is an interesting piece of work comprising isolation of and study of bioactivities of compounds from Psychotria nuda, comprising a very complete experimental part, an appropriate introduction and discussion part, with findings that are surely of interest in the field of phytochemistry and medicinal chemistry.
Therefore, I recommend the publication of this manuscript in Molecules.
The only minor corrections I suggest are the following:
- "Junior" in the authors' names should be corrected.
- "in vitro" should be italicized.
- "LPS" (lipopolysaccharide) should be defined in the text.
Author Response
Thank you for your suggestions. We took all your considerations into account.
Reviewer 2 Report
General comments
This study isolated and identified 19 compounds from leaves and twigs of Psychotria nuda. The anti-mycobacterial and anti-inflammatory activities of two of the compounds and a twin-compound mixture were examined. Although this study has some novelties, a few points still need to clarify.
Specific comments
1. This study isolated 19 compounds from leaves and twigs of P. nuda, but why only the mixture of compound (1 + 2) and compound 18, 19, not all ingredients, were used to test the bioactivities? Especially, the authors stress in the conclusion section that compound 9, 10, 11 and 14 were not published in the genus Psychotria, why the bioactivities of these compounds were not examined in this study? The authors should add the reason in Section 2.2.
2. Except the compounds examined, what about the anti-mycobacterial and anti-inflammatory activities of P. nuda extracts?
3. Line 56-63 should be deleted.
4. Line 183-185: Because this study did not investigate the anti-inflammatory activity of compound 11 and no paper reported its activities, alkaloid 11 should be removed from this sentence.
5. Line 215: The authors should state how compound 11 be separated and purified from the n-butanol fraction of P. nuda.
6. Line 256: Because Table 3 provides the IC50 value of L-NMMA, this data should be obtained from many data points that carried out at various concentrations of L-NMMA, the sentence “L-NMMA was used at 20 ug/mL” should be revised.
7. Some typing errors should be revised:
(1) Line 28: The first character of the word “5α-carboxystrictosidine” should be Italic.
(2) Line 105: The name “P. nuda” should be Italic.
(3) Line 151: The word ”RAM264.7” should be “RAW264.7”.
(4) Line 246, 254, 261: The microbial name should be Italic.
(5) Line 240, 252, 262: The order number should be superscript.
(6) Line 241, 251: The 2 in the word “CO2” should be subscript.
(7) References section should be carefully checked: (a) A few words in the titles of some references should not be Italic. (b) The dot should be added to the abbreviated journal names. (c) Some underlines should be removed.
Author Response
Dear reviewer,
Thank you for your cooperation. Basically, we tried to consider every aspect you pointed out.
Please find attached a file with the responses.

Reviewer 3 Report
In general, I do not like the manuscript: neither concept, neither performance. Some of the comments:
Lines 56-63 should be deleted.
Compounds should be bold (through the whole manuscript)
Coupling constant should be italic
Line 39: English: “for cure for the disease”
Choose mL or ml (do not mix)
Abstract: the last sentence: if this is the first report of the isolation of tryptamine derivative from a natural product it is not necessary to mention that this is the first report of the occurrence of the compound in P. nuda.
Table 2: incorrect name of rifampicin
Author Response
Dear reviewer,
We appreciate your suggestions and consider all the aspects you have pointed out.
Please find attached a file with our answers.

Reviewer 4 Report
Manuscript ID: molecules-449149
Type of manuscript: Article
Title: Antimycobacterial and Nitric Oxide Production Inhibitory Activities of
Triterpenes and Alkaloids from Psychotria nuda (Cham. & Schltdl.) Wawra
Authors: Almir Ribeiro de Carvalho Junior *, Rafaela Oliveira Ferreira,
Michel de Souza Passos, Samyra Imad da Silva Boeno, Lorena de Lima Glória
das Virgens, Thatiana Lopes Biá Ventura Simão, Sanderson Dias Calixto,
Elena Lassounskaia, Mario Geraldo De Carvalho, Raimundo Braz-Filho, Ivo Jose
Curcino Vieira
Nineteen known compounds have been isolated and identified from the leaves and twigs of Psychotria nuda, with their activity against Mycobacterium tuberculosis and inhibition of NO production evaluated. Some compounds showed activities in these assays, with IC50 values in the range 3.2–25.5 μg/mL. This manuscript reports bioactive components of Psychotria nuda, and some compounds showed interesting structures and bioactivities. Thus, it is recommended to be published in Molecules after major changes, as shown below and in the sticky notes of the attached manuscript pdf file.
1. Detailed the procedure of isolation of all compounds.
2. Demonstrate whether the isolation is completed under bioactivity guidance. If yes, which activity was followed by isolation, and which partitions and fractions were active.
3. Check carefully the references reported for all compounds (1–19) presented in this manuscript. If any data were not reported yet, present them in the manuscript.
Author Response
Dear reviewer,
We appreciate your suggestions and consider all the aspects you have pointed out.
Please find attached a file with our responses.

Round 2
Reviewer 3 Report
The authors impoved only few details I mentioned before.